# Uncertainties associated with GAN-generated datasets in high energy physics

Konstantin T. Matchev[1], Alexander Roman[1] and Prasanth Shyamsundar[1,2]⋆

**1** Institute for Fundamental Theory, Physics Department, University of Florida, Gainesville, FL 32611, USA
**2** Fermilab Quantum Institute, Fermi National Accelerator Laboratory, Batavia, IL 60510, USA

⋆ prasanth@fnal.gov

## Abstract

Recently, Generative Adversarial Networks (GANs) trained on samples of traditionally simulated collider events have been proposed as a way of generating larger simulated datasets at a reduced computational cost. In this paper we point out that data generated by a GAN cannot statistically be better than the data it was trained on, and critically examine the applicability of GANs in various situations, including a) for replacing the entire Monte Carlo pipeline or parts of it, and b) to produce datasets for usage in highly sensitive analyses or sub-optimal ones. We present our arguments using information theoretic demonstrations, a toy example, as well as in the form of a formal statement, and identify some potential valid uses of GANs in collider simulations.

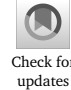

# 1 Introduction

Statistical inference in modern high energy experiments is typically done by comparing experimentally recorded data against simulated data corresponding to different theory models and parameters. In order to tap into the full statistical sensitivity offered by the experimental data, the simulated datasets need to keep up in volume to the real datasets, at least up to the same order of magnitude, and preferably beyond. This, coupled with the fact that Monte Carlo (MC) production, at least presently, is computationally expensive, poses a difficult challenge to the high energy community in terms of computational needs [1–4]. This problem will only be exacerbated at upcoming big-data experiments like the High Luminosity Large Hadron Collider (HL-LHC) [5].

Generative Adversarial Networks (GANs) [7] represent a class of machine learning (ML) models under which a generative network is trained to produce data that is statistically similar to a sample of training data. In the last few years, GANs and a few of their variants have been explored as tools to accelerate the MC production pipeline[1] in high energy physics [10–33]. The idea is to

1. Train a GAN on data simulated using the available, slow, MC pipeline. Alternatively, the GAN can be trained on real calibration data samples as done in [10].

2. Use the trained GAN to generate additional, statistically similar, data.

GANs can be several orders of magnitude faster than the simulators they are trying to mimic, and this speed-up can potentially be leveraged to bring down the overall computational cost of the data generation process [10–33]. Additionally, by combining several event generation stages together and directly generating high-level features on which analyses will be performed, GANs can also potentially reduce the MC disk space requirements [29].

There are several works in the literature that propose to replace specific steps of the high energy MC generation pipeline with GANs. For example, the usage of GANs to *only* perform

---

[1]Unless specified otherwise, in what follows we shall take "event generation" to mean the complete MC production chain in high energy physics [8], including the stages of parton-level event generation, fragmentation and hadronization, detector and electronics simulation, and object reconstruction.

detector/calorimeter simulation was explored in [10–23]. The use of GANs for simulating the underlying hard process was considered in [24–26], while the use of GANs for pileup description was explored in [27, 28]. Recent works have also pushed the idea of replacing the *entire* reconstructed-event generation pipeline with a GAN [23, 24, 27–32]. Each of these applications of GANs would differ in the nature of the training data used.

In what follows, the term "true" simulator/generator will refer to the method used to produce the data the GAN will be trained on. Note that the training data still needs to be generated using the non-GAN true simulator. Consequently, in order to actually exploit the advantages offered by a GAN, its training dataset needs to be much smaller than the number of simulated events required by the analysis. A natural question, then, is what is the actual benefit (in a statistical sense) from using this enlarged dataset.

In this paper, we present an argument cautioning against the usage of GANs to meet the simulation requirements of an analysis. The basic idea is that any feature that cannot be resolved using the training data, cannot be extrapolated from the training data. As an analogy, if a grainy, low-resolution CCTV image is sufficient to identify a person on the image, then one can trust a software enhancement [34] that reduces the noise and increases the image-resolution to reveal the person's face more clearly. On the other hand, if the low-res image is consistent with two or more people, even after taking all the available information in the raw image file and noise models into account, then one cannot resolve the ambiguity through software enhancement. The very fact that there are multiple potential matches at the low-res level implies that one can create plausible image-enhancers which can make the synthetic high-res image point to any of the potential matches.

As an added impetus for the reader to take our point seriously, we note that if one entertains the idea of using a GAN to replace the entire MC production pipeline, then one should also seriously consider the possibility of training a GAN on experimental data to artificially boost the luminosity of an experiment, essentially producing additional synthetic experimental data at significantly reduced costs.

Since a) the justification for leveraging a GAN for speed-up is that the training dataset, by itself, is not large enough to meet the simulation requirements, and b) GANs only extrapolate from the training dataset, we claim that GANs cannot be used to speed-up any sequence of steps of the event generation pipeline which is a bottleneck on the overall *uncertainty* from the MC dataset. In the rest of this paper, we present this argument carefully and examine the applicability of GANs in various situations. In Sections 2–4, we focus on using GANs to replace the entire Monte Carlo pipeline, while in Section 7 we consider the possibility of replacing only certain parts of it. In Section 5 we discuss the possibility of amplifying datasets using GANs, as suggested in [33]. In Section 6, we reconcile our results with those of previous work, and explain the seeming contradictions. In Section 7, we identify some potential uses of GANs whose validity is not affected by our arguments. We summarize in Section 8. In Appendix A we present the results from a toy example for illustration purposes.

## 2  Formulation of the statement

Let us begin by considering the situation where a GAN will be used to replace the entire MC production pipeline. In that case, our main argument can be formalized as follows:

**Statement 1.** Under the following two assumptions, the model (or model parameter) discriminating power achievable using only the available true-simulated data cannot be improved by augmenting the true-simulated data with additional GAN-generated data.

**Assumptions:**

1. The GAN will be used to circumvent the MC pipeline, starting from the parton-level event generation and going all the way up to either detector hits or reconstructed high-level objects.

2. There are no restrictions on the types of analyses a practitioner is allowed to perform using the real and the (GAN/true) simulated data. For example, the practitioner should not be restricted only to a particular technique, which may or may not be optimal in its usage of the available data.

$$\diamond$$

The statement holds for *any* agreed-upon evaluation metric for capturing the model discriminating power achieved when performing any of the candidate analyses (available to the practitioner) on the available data. Note that this statement is stronger than the statement that a sample of $N$ events generated using a GAN trained on a finite sample of $N_{\text{train}}$ true-simulated events will be *asymptotically* (i.e., in the large $N$ limit, but with fixed $N_{\text{train}}$) distinguishable from data generated using the true simulator.

Apart from its two listed assumptions, Statement 1 does not rely on any additional assumptions, and holds in a variety of situations. In particular,

1. To be maximally conservative, we will ignore any deficiencies in the training of a GAN, and allow the data generated by the GAN to agree with the training data on every conceivable distribution, up to statistical fluctuations. Note that this is an idealized situation which actually bolsters the case for GANs.

2. We will not make any assumptions about the region of phase-space in which GANs will be used to provide additional simulated data. In other words, our argument holds regardless of whether the GAN will be used to sample events from the "bulk" of a distribution (well represented in the training data) or the "tail" of a distribution (poorly represented in the training data).

The proof of Statement 1 is presented in Section 4. We will develop the intuition for the proof in Section 3.

## 3 Quantitative discussion

### 3.1 Why simulate more events?

It is instructive to review the primary role played by simulated data in collider analyses. The goal of the collider experiments is to probe the underlying true theory of nature and figure out which of the competing hypotheses (models/model parameter values) best matches it[2]. The only handle we have on the true theory of nature is the real experimental data. This is illustrated in the right side of Figure 1. On the other hand, since our theoretical models do not provide closed form expressions for the distributions of experimental data (especially once we account for the selection cuts, efficiencies and detector resolution), simulated data serve as our respective handle on the theoretical model candidates, as illustrated in the left side of Figure 1. The more real experimental events we have, the better our handle on the true theory. Similarly, the more simulated data we have, the better our handle on the competing theory models [35–39]. The quality of the hypothesis test or parameter measurement depends on the

---

[2]In the case of unmodeled searches, the goal is to identify deviations from the null hypothesis, but this is a moot detail.

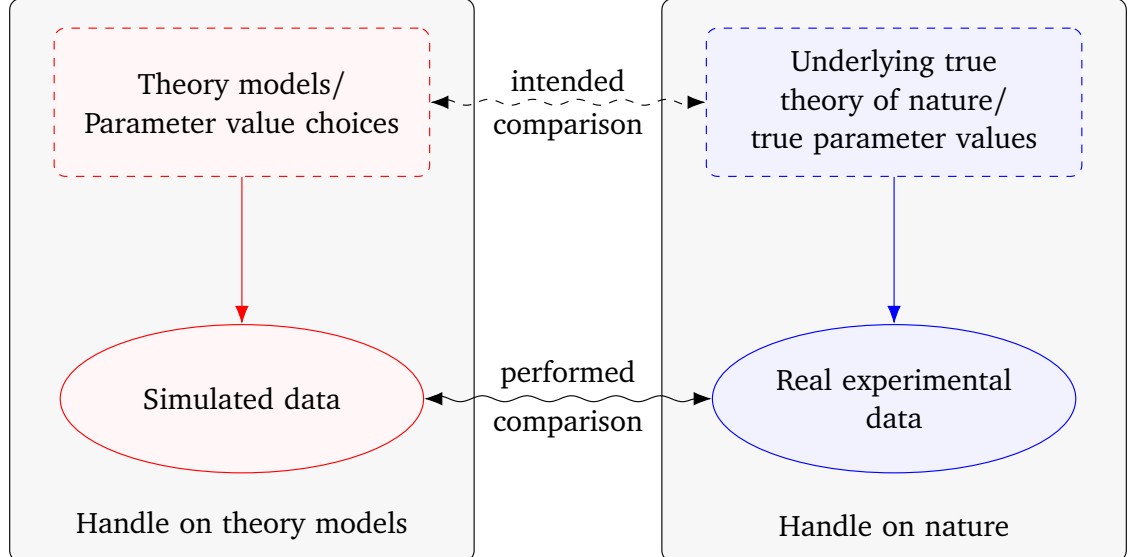

Figure 1: A diagram illustrating the simulation-based inference paradigm used in high-energy physics for testing theory models against experimental data.

quality of both these handles, and hence the need for simulated data to keep up in volume with real data. Note that this inference paradigm, under which experimental high energy physics works, requires that the simulated data be faithful to the assumed theoretical hypotheses.[3]

In principle, as we increase the amount of simulated (and real) data, the statistical uncertainties in the estimated theoretical (and true) distributions decrease. This allows us to better distinguish between models and model parameter values with similar (but not identical) predictions for the data distribution. The central question here is whether the inclusion of GAN-generated data leads to the same statistical benefit. In this section we will focus on the quality of the handle on the theory model provided by the true-simulated and GAN-generated datasets, i.e, the vertical red arrow in the left side of Figure 1. In particular, we will demonstrate that GAN-generated data cannot contain any more information about the underlying theory model than the true-simulated training data it is based on.

## 3.2 Information theoretic demonstrations

Let $\theta$ represent the model parameters being measured in a parameter measurement analysis. In the context of hypothesis testing analyses, $\theta$ can capture the choice of model as well as the associated parameters. The production of GAN-generated data typically proceeds in the following three steps:

1. A value of $\theta$ is chosen, and a training dataset $D_{\text{train}}$ is generated using the true simulator for the chosen value of $\theta$. Let the probability distribution of the training dataset conditional on $\theta$ be given by $\mathcal{P}(D_{\text{train}} \mid \theta)$.

2. A GAN is trained on the full training dataset $D_{\text{train}}$. Let $\text{GAN}_{\text{trained}}$ represent the trained GAN. Let $\mathcal{P}(\text{GAN}_{\text{trained}} \mid D_{\text{train}})$ be the probability distribution of the trained GAN (in the space of possible network architectures and parameters), conditional on the training data. This way, we allow for $\text{GAN}_{\text{trained}}$ to not necessarily be fully determined by

---

[3]A moot point here is that there are different aspects of the generative models that we are uncertain about, some of which we want to be sensitive to (e.g., parton level theory), and others we want to minimize impact from (e.g., detector effects, or non-perturbative QCD in BSM physics searches). These are treated on different footings in Monte Carlo research.

the training data—typically there is some stochasticity involved in the network training process, e.g., in the initialization of the network weights prior to the training, the choice of network architectures, etc.

3. Finally, the trained network $\text{GAN}_{\text{trained}}$ is used to generate the dataset $D_{\text{GAN}}$. Let $\mathcal{P}(D_{\text{GAN}} \mid \text{GAN}_{\text{trained}})$ be the probability density of the entire GAN-generated dataset $D_{\text{GAN}}$, conditional on the trained GAN, $\text{GAN}_{\text{trained}}$.

Due to the Markovian nature of the production of $D_{\text{GAN}}$, via $D_{\text{train}}$ and $\text{GAN}_{\text{trained}}$, we can write the joint distribution of $(D_{\text{GAN}}, \text{GAN}_{\text{trained}}, D_{\text{train}})$ conditional on $\boldsymbol{\theta}$ as

$$
\begin{aligned}
\mathcal{P}(D_{\text{GAN}}, \text{GAN}_{\text{trained}}, D_{\text{train}} \mid \boldsymbol{\theta}) = {} & \mathcal{P}(D_{\text{train}} \mid \boldsymbol{\theta}) \\
& \times \mathcal{P}(\text{GAN}_{\text{trained}} \mid D_{\text{train}}) \\
& \times \mathcal{P}(D_{\text{GAN}} \mid \text{GAN}_{\text{trained}}).
\end{aligned}
\tag{1}
$$

This means that the likelihood ratio of $(D_{\text{GAN}}, \text{GAN}_{\text{trained}}, D_{\text{train}})$ under two different values of $\boldsymbol{\theta}$, say $\boldsymbol{\theta}_0$ and $\boldsymbol{\theta}_1$, is simply given by

$$
\begin{aligned}
& \frac{\mathcal{P}(D_{\text{GAN}}, \text{GAN}_{\text{trained}}, D_{\text{train}} \mid \boldsymbol{\theta}_0)}{\mathcal{P}(D_{\text{GAN}}, \text{GAN}_{\text{trained}}, D_{\text{train}} \mid \boldsymbol{\theta}_1)} \\
& \quad = \frac{\mathcal{P}(D_{\text{train}} \mid \boldsymbol{\theta_0}) \times \mathcal{P}(\text{GAN}_{\text{trained}} \mid D_{\text{train}}) \times \mathcal{P}(D_{\text{GAN}} \mid \text{GAN}_{\text{trained}})}{\mathcal{P}(D_{\text{train}} \mid \boldsymbol{\theta_1}) \times \mathcal{P}(\text{GAN}_{\text{trained}} \mid D_{\text{train}}) \times \mathcal{P}(D_{\text{GAN}} \mid \text{GAN}_{\text{trained}})}
\end{aligned}
\tag{2a}
$$

$$
\quad = \frac{\mathcal{P}(D_{\text{train}} \mid \boldsymbol{\theta}_0)}{\mathcal{P}(D_{\text{train}} \mid \boldsymbol{\theta}_1)}.
\tag{2b}
$$

If $\boldsymbol{\theta}$ is a continuous parameter, from (1) we can also write the score function $\partial_{\boldsymbol{\theta}} \mathcal{P} / \mathcal{P}$ for the observable $(D_{\text{GAN}}, \text{GAN}_{\text{trained}}, D_{\text{train}})$ as

$$
\frac{\partial_{\boldsymbol{\theta}} \mathcal{P}(D_{\text{GAN}}, \text{GAN}_{\text{trained}}, D_{\text{train}} \mid \boldsymbol{\theta})}{\mathcal{P}(D_{\text{GAN}}, \text{GAN}_{\text{trained}}, D_{\text{train}} \mid \boldsymbol{\theta})} = \frac{\partial_{\boldsymbol{\theta}} \mathcal{P}(D_{\text{train}} \mid \boldsymbol{\theta})}{\mathcal{P}(D_{\text{train}} \mid \boldsymbol{\theta})}.
\tag{3}
$$

Note that the likelihood ratio and the score function of $(D_{\text{GAN}}, \text{GAN}_{\text{trained}}, D_{\text{train}})$ in (2b) and (3), respectively, are in fact both independent of the trained GAN and the GAN-generated data. In Sections 3.2.1–3.2.3, we will use this property to demonstrate, using information theoretic concepts, that the GAN does not provide any additional information on $\boldsymbol{\theta}$, beyond what is contained in the training dataset $D_{\text{train}}$.

### 3.2.1 Mutual information

The amount of information contained in a simulated dataset $D$ about the underlying $\boldsymbol{\theta}$ can be captured by their mutual information. Let $\mathcal{I}^{\text{mutual}}(X \, ; \, Y)$ be the mutual information of $X$ and $Y$. Here, we will show that $\mathcal{I}^{\text{mutual}}(\boldsymbol{\theta} \, ; \, D_{\text{GAN}}) \leq \mathcal{I}^{\text{mutual}}(\boldsymbol{\theta} \, ; \, D_{\text{train}})$, i.e., that $D_{\text{GAN}}$ cannot contain any more information about $\boldsymbol{\theta}$ than $D_{\text{train}}$.

$\mathcal{I}^{\text{mutual}}(X \, ; \, Y)$ can be written as

$$
\mathcal{I}^{\text{mutual}}(X \, ; \, Y) \equiv \mathcal{I}^{\text{mutual}}(Y \, ; \, X) \equiv \mathcal{H}(X) - \mathcal{H}(X \mid Y),
\tag{4}
$$

where $\mathcal{H}(X)$ is the entropy[4] of $X$ and $\mathcal{H}(X \mid Y)$ is the conditional entropy[4] of $X$ given $Y$.[5] Using the chain rule for conditional entropy given by

$$
\mathcal{H}(X, Y) = \mathcal{H}(X) + \mathcal{H}(Y \mid X),
\tag{5}
$$

---

[4]If $X$ is a continuous variable, $\mathcal{H}$ denotes the differential entropy or the conditional differential entropy.

[5]In discussing the information, entropy, and conditional entropy of $\boldsymbol{\theta}$, we are implicitly assuming the existence of a prior distribution for $\boldsymbol{\theta}$. The results derived in this section are independent of the exact choice of this prior distribution.

we have

$$\mathcal{H}(D_{\text{GAN}}, \text{GAN}_{\text{trained}}, D_{\text{train}}, \boldsymbol{\theta}) = \mathcal{H}(D_{\text{train}}) + \mathcal{H}(D_{\text{GAN}}, \text{GAN}_{\text{trained}} \mid D_{\text{train}}) \\ + \mathcal{H}(\boldsymbol{\theta} \mid D_{\text{GAN}}, \text{GAN}_{\text{trained}}, D_{\text{train}}). \tag{6}$$

Applying the chain rule in a different way, we can also write

$$\mathcal{H}(D_{\text{GAN}}, \text{GAN}_{\text{trained}}, D_{\text{train}}, \boldsymbol{\theta}) = \mathcal{H}(D_{\text{train}}) + \mathcal{H}(\boldsymbol{\theta} \mid D_{\text{train}}) \\ + \mathcal{H}(D_{\text{GAN}}, \text{GAN}_{\text{trained}} \mid D_{\text{train}}, \boldsymbol{\theta}). \tag{7}$$

From (6) and (7), we have

$$\mathcal{H}(D_{\text{GAN}}, \text{GAN}_{\text{trained}} \mid D_{\text{train}}) + \mathcal{H}(\boldsymbol{\theta} \mid D_{\text{GAN}}, \text{GAN}_{\text{trained}}, D_{\text{train}}) = \\ \mathcal{H}(D_{\text{GAN}}, \text{GAN}_{\text{trained}} \mid D_{\text{train}}, \boldsymbol{\theta}) + \mathcal{H}(\boldsymbol{\theta} \mid D_{\text{train}}). \tag{8}$$

On the other hand, we can also write

$$\mathcal{P}(D_{\text{GAN}}, \text{GAN}_{\text{trained}} \mid D_{\text{train}}, \boldsymbol{\theta}) \equiv \frac{\mathcal{P}(D_{\text{GAN}}, \text{GAN}_{\text{trained}}, D_{\text{train}} \mid \boldsymbol{\theta})}{\mathcal{P}(D_{\text{train}} \mid \boldsymbol{\theta})} \tag{9a}$$

$$= \mathcal{P}(\text{GAN}_{\text{trained}} \mid D_{\text{train}}) \times \mathcal{P}(D_{\text{GAN}} \mid \text{GAN}_{\text{trained}}), \tag{9b}$$

where to arrive at the second line we have used (1). Using the Markovian nature of the production of $D_{\text{GAN}}$, we can rewrite (9b) as

$$\mathcal{P}(D_{\text{GAN}}, \text{GAN}_{\text{trained}} \mid D_{\text{train}}, \boldsymbol{\theta}) = \mathcal{P}(D_{\text{GAN}}, \text{GAN}_{\text{trained}} \mid D_{\text{train}}), \tag{10}$$

and therefore

$$\mathcal{H}(D_{\text{GAN}}, \text{GAN}_{\text{trained}} \mid D_{\text{train}}, \boldsymbol{\theta}) = \mathcal{H}(D_{\text{GAN}}, \text{GAN}_{\text{trained}} \mid D_{\text{train}}). \tag{11}$$

The last relation allows us to cancel the first term on each side of (8), simplifying it to

$$\mathcal{H}(\boldsymbol{\theta} \mid D_{\text{GAN}}, \text{GAN}_{\text{trained}}, D_{\text{train}}) = \mathcal{H}(\boldsymbol{\theta} \mid D_{\text{train}}). \tag{12}$$

Furthermore, since $\mathcal{H}(\boldsymbol{\theta} \mid D_{\text{GAN}}) \geq \mathcal{H}(\boldsymbol{\theta} \mid D_{\text{GAN}}, \text{GAN}_{\text{trained}}, D_{\text{train}})$, we have

$$\mathcal{H}(\boldsymbol{\theta} \mid D_{\text{GAN}}) \geq \mathcal{H}(\boldsymbol{\theta} \mid D_{\text{train}}). \tag{13}$$

Using (4), this can be rewritten as

$$\mathcal{I}^{\text{mutual}}(\boldsymbol{\theta} \; ; \; D_{\text{GAN}}) \leq \mathcal{I}^{\text{mutual}}(\boldsymbol{\theta} \; ; \; D_{\text{train}}), \tag{14}$$

which shows that $D_{\text{GAN}}$ cannot contain any more information on $\boldsymbol{\theta}$ than $D_{\text{train}}$.

### 3.2.2 Kullback–Leibler divergence between two values of $\theta$

The distinguishability of two hypotheses represented by $\boldsymbol{\theta_0}$ and $\boldsymbol{\theta_1}$ is captured by the Kullback–Leibler (KL) divergence $\mathcal{D}^{\text{KL}}$ between the two respective distributions. In this section, we will compare the distinguishability of the hypotheses a) using only the training dataset $D_{\text{train}}$ and b) using $(D_{\text{GAN}}, \text{GAN}_{\text{trained}}, D_{\text{train}})$.

$$\mathcal{D}^{\text{KL}}_{(D_{\text{GAN}}, \text{GAN}_{\text{trained}}, D_{\text{train}})}(\boldsymbol{\theta_0} \| \boldsymbol{\theta_1}) = \int d(D_{\text{GAN}}, \text{GAN}_{\text{trained}}, D_{\text{train}}) \\ \mathcal{P}(D_{\text{GAN}}, \text{GAN}_{\text{trained}}, D_{\text{train}} \mid \boldsymbol{\theta_0}) \\ \log \left[ \frac{\mathcal{P}(D_{\text{GAN}}, \text{GAN}_{\text{trained}}, D_{\text{train}} \mid \boldsymbol{\theta_0})}{\mathcal{P}(D_{\text{GAN}}, \text{GAN}_{\text{trained}}, D_{\text{train}} \mid \boldsymbol{\theta_1})} \right], \tag{15}$$

where the integral is over the space of datasets $D_{\text{GAN}}$ and $D_{\text{train}}$, and the space of GAN-instances $\text{GAN}_{\text{trained}}$, with an appropriate integration measure. Using (2b), and integrating out $D_{\text{GAN}}$ and $\text{GAN}_{\text{trained}}$, we can write this as

$$\mathcal{D}^{\text{KL}}_{(D_{\text{GAN}},\text{GAN}_{\text{trained}},D_{\text{train}})}(\boldsymbol{\theta}_0 \,\|\, \boldsymbol{\theta}_1) = \int dD_{\text{train}} \, \mathcal{P}(D_{\text{train}} \mid \boldsymbol{\theta}_0) \, \log\left[\frac{\mathcal{P}(D_{\text{train}} \mid \boldsymbol{\theta}_0)}{\mathcal{P}(D_{\text{train}} \mid \boldsymbol{\theta}_1)}\right] \tag{16a}$$

$$\equiv \mathcal{D}^{\text{KL}}_{D_{\text{train}}}(\boldsymbol{\theta}_0 \,\|\, \boldsymbol{\theta}_1). \tag{16b}$$

Furthermore, from the data processing inequality for $f$-divergences, we have

$$\mathcal{D}^{\text{KL}}_{D_{\text{GAN}}}(\boldsymbol{\theta}_0 \,\|\, \boldsymbol{\theta}_1) \leq \mathcal{D}^{\text{KL}}_{(D_{\text{GAN}},\text{GAN}_{\text{trained}},D_{\text{train}})}(\boldsymbol{\theta}_0 \,\|\, \boldsymbol{\theta}_1). \tag{17}$$

From (16) and (17) we can deduce that

$$\mathcal{D}^{\text{KL}}_{D_{\text{GAN}}}(\boldsymbol{\theta}_0 \,\|\, \boldsymbol{\theta}_1) \leq \mathcal{D}^{\text{KL}}_{D_{\text{train}}}(\boldsymbol{\theta}_0 \,\|\, \boldsymbol{\theta}_1). \tag{18}$$

This result shows that the two hypotheses ($\boldsymbol{\theta} = \boldsymbol{\theta}_0$ versus $\boldsymbol{\theta} = \boldsymbol{\theta}_1$) are no more distinguishable using GAN-generated data than when using only the training data.

### 3.2.3 Fisher information in simulated data about the model parameter

If $\boldsymbol{\theta}$ is a continuous parameter, the amount of information contained in the GAN-generated data about the value of $\boldsymbol{\theta}$ used in the simulation can be captured by the Fisher information [40]. Let $\mathcal{I}^{\text{Fisher}}_X(\boldsymbol{\theta})$ be the Fisher information in the observable $X$, where $X$ stands for the dataset of interest. The Fisher information in the training data is given by

$$\mathcal{I}^{\text{Fisher}}_{D_{\text{train}}}(\boldsymbol{\theta}) = \int dD_{\text{train}} \, \mathcal{P}(D_{\text{train}} \mid \boldsymbol{\theta}) \left[\frac{\partial_{\boldsymbol{\theta}} \mathcal{P}(D_{\text{train}} \mid \boldsymbol{\theta})}{\mathcal{P}(D_{\text{train}} \mid \boldsymbol{\theta})}\right]^2, \tag{19}$$

where the integral is taken over the space of possible instances of $D_{\text{train}}$. Now let us turn to the Fisher information contained in $(D_{\text{GAN}},\text{GAN}_{\text{trained}},D_{\text{train}})$. Using (3), we can write

$$\mathcal{I}^{\text{Fisher}}_{(D_{\text{GAN}},\text{GAN}_{\text{trained}},D_{\text{train}})}(\boldsymbol{\theta}) = \int d(D_{\text{GAN}},\text{GAN}_{\text{trained}},D_{\text{train}})$$
$$\mathcal{P}(D_{\text{GAN}},\text{GAN}_{\text{trained}},D_{\text{train}} \mid \boldsymbol{\theta}) \left[\frac{\partial_{\boldsymbol{\theta}} \mathcal{P}(D_{\text{train}} \mid \boldsymbol{\theta})}{\mathcal{P}(D_{\text{train}} \mid \boldsymbol{\theta})}\right]^2. \tag{20}$$

We can now integrate out $D_{\text{GAN}}$ and $\text{GAN}_{\text{trained}}$ to get

$$\mathcal{I}^{\text{Fisher}}_{(D_{\text{GAN}},\text{GAN}_{\text{trained}},D_{\text{train}})}(\boldsymbol{\theta}) = \int dD_{\text{train}} \, \mathcal{P}(D_{\text{train}} \mid \boldsymbol{\theta}) \left[\frac{\partial_{\boldsymbol{\theta}} \mathcal{P}(D_{\text{train}} \mid \boldsymbol{\theta})}{\mathcal{P}(D_{\text{train}} \mid \boldsymbol{\theta})}\right]^2 \tag{21a}$$

$$\equiv \mathcal{I}^{\text{Fisher}}_{D_{\text{train}}}(\boldsymbol{\theta}). \tag{21b}$$

This shows that $(D_{\text{GAN}},\text{GAN}_{\text{trained}},D_{\text{train}})$ contains no more Fisher information than $D_{\text{train}}$. Furthermore, from the data processing inequality [40] we have

$$\mathcal{I}^{\text{Fisher}}_{D_{\text{GAN}}}(\boldsymbol{\theta}) \leq \mathcal{I}^{\text{Fisher}}_{(D_{\text{GAN}},\text{GAN}_{\text{trained}},D_{\text{train}})}(\boldsymbol{\theta}). \tag{22}$$

From (21) and (22), we have

$$\mathcal{I}^{\text{Fisher}}_{D_{\text{GAN}}}(\boldsymbol{\theta}) \leq \mathcal{I}^{\text{Fisher}}_{D_{\text{train}}}(\boldsymbol{\theta}). \tag{23}$$

This result is in agreement with, and thus reaffirms the conclusions of, (14) and (18), only this time in terms of Fisher information.

## 3.3 Intuition behind the math

The results in (14), (18), and (23) can be intuitively understood as follows. Let us say we need a sample of events from a diagram involving the top quark. We set the mass of the top quark in the true simulator to 173 GeV and produce $N_{\text{train}}$ number of events. For the sake of argument let us assume that, within the statistical fluctuations, the produced dataset is 'similarly' consistent with both $m_{\text{top}} = 173$ GeV and $m_{\text{top}} = 175$ GeV. Loosely speaking, in an alternative universe, running the simulator with $m_{\text{top}}$ set to 175 GeV could produce the exact same dataset without raising any alarms.

Now let us say we use these $N_{\text{train}}$ simulated events as training events to produce additional GAN-generated events. The combined dataset consisting of both the training events and the GAN-generated events will continue to be 'similarly' consistent with both 173 GeV and 175 GeV. This is because the training data on which the GAN is trained, could have been from either of the two mass values—this ambiguity will be frozen into the GAN-generated dataset, regardless of the number of GAN-generated events used by the analysis.

In technical terms, the statistical uncertainties in the true-simulated training data (depicted by the green dashed line in Figure 2) are inherited by GAN-generated data as systematic uncertainties[6] (the blue dotted line in Figure 2). This is because the specific realization of the training data can be thought of as a probabilistic nuisance parameter in the generative model of the trained GAN. Additionally, the systematic uncertainties of the GAN-generated data will also contain the systematic uncertainties in the training data (the red dotted line) and the ML training related uncertainties (which we are neglecting here). Since systematic uncertainties do not reduce with increasing sample size, a GAN cannot be used to beat down the statistical uncertainties in the data it was trained on—note how the solid blue line in Figure 2 representing the total GAN uncertainty asymptotes *not* to the asymptotic value of the total true simulation uncertainty, but only to the value corresponding to the finite fixed number $N_{\text{train}}$ of training events.

The qualitative behavior exhibited in Figure 2 is demonstrated quantitatively with a toy example in Appendix A, where we consider the task of estimating the mean of a normal distribution using a finite sample of events generated from a true normal distribution, as well as a number of events sampled from a learned distribution. We show that the ultimate precision is limited by the finiteness of the training sample and does not improve by producing additional events from the learned distribution. On the other hand, if we increase the number of "true simulated" events in the dataset, the precision does improve, as expected.

## 3.4 Addressing the interpolation argument

A common argument used to justify the usage of GANs for event generation is that GANs can interpolate and learn approximate functional forms of the underlying distribution using the available training data [33]. In this context, it is important to remember the causal relationship between the size of a simulated dataset and the "smoothness" of a histogram constructed from the dataset.

1. As the number of true-simulated events increases, the histogram constructed from the events gets closer to the underlying true distribution.

2. In high energy physics, the underlying distributions of event-attributes are typically smooth. Statistical fluctuations on the other hand tend to be "noisy".

---

[6]By statistical and systematic "uncertainties in/from/of simulated data", we mean the uncertainties in a particular analysis caused, respectively, by the limited statistics of the simulated data and the uncertainties in the generative model. Note that experiments usually include *all* uncertainties from simulated data under systematic uncertainties, with the label "statistical uncertainties" in the experimental results being reserved for statistical uncertainties from real experimental data.

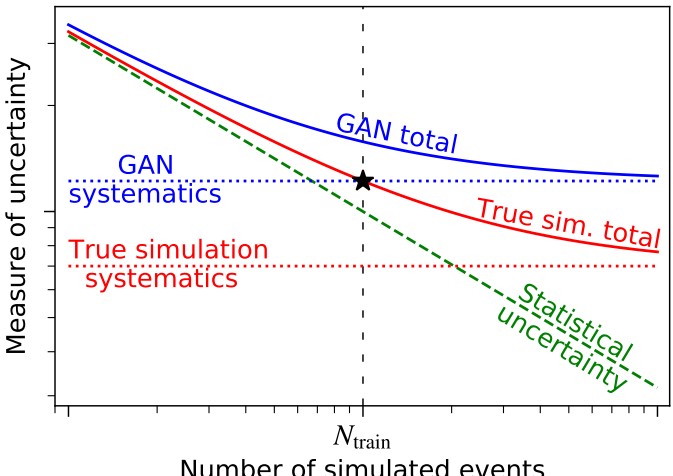

Figure 2: Typical evolution of various uncertainties with the number of simulated events. The green dashed line corresponds to the statistical uncertainty from simulation, while the red dotted and blue dotted lines represent the systematic uncertainty of the true simulation and the GAN simulation, respectively. The red (blue) solid line gives the total uncertainty of the true simulation (the GAN simulation), with the corresponding statistical and systematic uncertainties added in quadrature. We assume that the GAN is trained on a sample of $N_{\text{train}}$ events, and as a result, the GAN-generated data inherits the total uncertainty from the training sample as depicted by the ★ symbol. For concreteness, the blue curves correspond to purely GAN-generated data, and not GAN-augmented data.

3. As a result, increasing the simulation statistics leads to smoother histograms.

The causal flowchart is given by

$$\text{More events} \rightarrow \text{histogram approaches true distribution} \rightarrow \text{histogram smoothens out} \quad (24)$$

Note that the first point on the list above and the first arrow in (24) hold only because the true simulated events are consistent with the underlying distribution of the true simulator.

Interpolating distributions to generate more simulated events seeks to reverse the causality in (24). Roughly, the approach can be thought of as a) smooth out a histogram to get closer to the true distribution, and b) produce more events from the smoothed distribution. Unfortunately, smoothing out a histogram will not lead to the true underlying distribution. Rather, it will lead to one of several smooth distributions consistent with the data the interpolation is performed on. In low-dimensional data, this is reminiscent of the availability of different choices of smoothing algorithms, e.g., kernel density estimation (KDE), and parameters within the algorithms, e.g. choice of kernel in KDE. Several different choices for the smoothing prescription can lead to similarly valid interpolations. However, none of these interpolations is objectively better than the others in all situations, and none of them is guaranteed to match the true underlying distribution better than the others.

For concreteness, in our top-quark example in Section 3.3, let us say an interpolation is performed on the training data simulated with $m_{\text{top}}$ set to 173 GeV. Recall that the training data was consistent with $m_{\text{top}} = 175$ GeV as well. Because of this, the interpolated distribution will not be able to (and should not be used to) definitively distinguish between 173 GeV and 175 GeV—properly accounting for the interpolation uncertainties will ensure of that.

**We are not claiming here that techniques like interpolation, function approximation, curve fitting, etc. are not useful in data analysis in physics.** In fact, many HEP analyses

use these techniques extensively, see e.g. [41], to improve their sensitivities. The crucial point, however, is that these techniques are (and can always be) used at the analysis stage, and not the MC production stage.[7] The reason for requiring more simulated events is to probe the underlying theory model distribution (the red block in Fig.1) better than it can be done with interpolations. The interpolation techniques exploit our knowledge that the underlying true distribution must be smooth up to certain scales or resolutions, and allow us to utilize the information already contained in the available data (real and simulated) efficiently in our analyses. But a unified message from statistics and information theoretic concepts like Fisher information is that there is a limit to the amount of information that can be extracted from a finite sample of datapoints. The only way to overcome this limitation is by producing more real and simulated events, and this is precisely the reason why future high-luminosity HEP experiments require large simulation statistics. As we have demonstrated, GANs trained on a finite sample of true-simulated data cannot be used to satisfy this need.

## 4 Proof of Statement 1

### 4.1 Challenges

Proving Statement 1 poses a couple of challenges. We will first present these challenges, before proceeding to the proof.

- In Section 3.2, we demonstrated how GAN-generated data will not contain any more information than the training data about the underlying theory model used to generate the data. For this purpose, we used information theoretic concepts like mutual information, KL divergence, and Fisher Information. As discussed earlier, these metrics capture the quality of the simulation-handle on our theory models—**the left side of Figure 1**.

  While the left side of Figure 1 is the primary focus of this paper, in general the sensitivity of a simulation-based analysis will depend both on the simulation-handle on the theory models (left side of Figure 1) and on the real-data-handle on nature (right side of Figure 1). Unfortunately, there are no standard measures in the literature for capturing the overall sensitivity *achievable* in such analyses.[8]

- One can introduce variations in the GAN-training prescription which break the Markovian structure captured by (1). For example, instead of training different GANs for different choices of $\theta$, one can train a single network simultaneously on data from several different $\theta$-choices. Once trained, such networks, referred to as conditional GANs (cGANs) [42], can (approximately) mimic data from any choice of $\theta$, which is provided as a parameter to the network. Another variation is to include some domain knowledge about the underlying simulation to improve the GAN-training [25].

  The information theoretic demonstrations in Section 3.2 will have to be modified to accommodate each of these alternative GAN-training approaches. Instead, here we will provide a direct proof of Statement 1 to cover all cases.

---

[7]Note that interpolation techniques are also used in building our true-simulation models, for example, in the determination of parton distribution functions. This however is a moot point, and not pertinent to our current discussion.

[8]A generalization of Fisher information to simulation-based analyses is presented in [39]. However, this generalization does not have an associated result analogous to Cramér–Rao bound for the standard Fisher information.

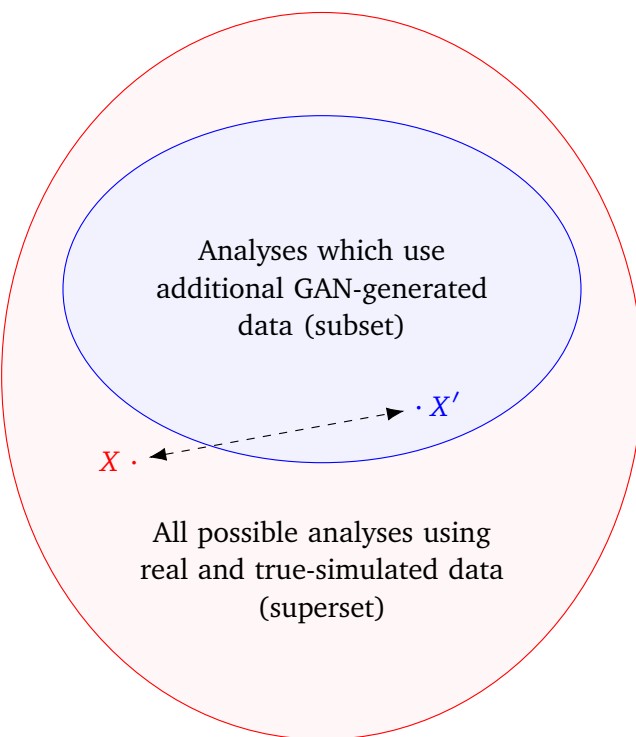

Figure 3: A Venn diagram illustrating the proof of Statement 1 discussion in Section 4.2

.

## 4.2 The proof

The proof of Statement 1 lies in recognizing that it is a tautology under the assumptions it is conditional on: One of the assumptions required for Statement 1 is that the practitioners who analyze the real and simulated data should be free to use any analysis technique.

In particular, if provided with some true-simulated data, the practitioners can choose, *as part of their analysis*, to train a GAN and use additional simulated data from it. So, using GANs to produce additional simulated data can be viewed as an *analysis technique* which uses only the true-simulated data, as opposed to a data generation technique. This implies that the model discriminating power **achievable** using only the available true-simulated data cannot be improved by augmenting the true-simulated data with additional GAN-generated data, thereby completing the proof of Statement 1.

The proof is illustrated as a Venn diagram in Figure 3, where the set of analyses which use additional GAN-generated data is a subset of all possible analyses. The maximal sensitivity achievable within the subset cannot be better than the overall maximal sensitivity. This proof captures the following essence of the various data processing inequalities used in Section 3.2: *The best that can be done with some given data, by definition, cannot be improved upon by processing the data further.*

## 4.3 Discussion

The nature of the proof might suggest that while technically correct, Statement 1 is not of practical relevance. After all, it should not matter if the GANs are used at the simulation stage or the analysis stage, as long as it improves the sensitivity of the analysis. However, Statement 1 is in fact practically relevant.

The point is that HEP analyses can, and already do, use sophisticated techniques to in-

terpolate from, and approximate the distribution of, the available simulated data, e.g. see Appendix B of [41]. When using these techniques, one typically takes into account the known limitations of the generative model used by the true-simulator in order to perform the interpolations *safely*. Furthermore, HEP analyses have been advancing towards the upper-limit on sensitivity set by the Fisher information, with techniques like Matrix Element Method and its variants [43–45] as well as ML-based techniques [46–49].

When a sensitive analysis (which incorporates data interpolation) reports a simulation requirement of $N$ events, we need to provide $N$ events from the underlying theory distribution. If GAN-generated data is used instead, the analysis will be **interpolating from an interpolation**. This will either lead to no improvement in sensitivity if the uncertainties are properly handled, or lead to incorrect results if the uncertainties are not properly handled.

The argument here hinges on the analysis performed using the real and simulated data being close to optimal. A natural question that arises is whether GAN-generated data can help improve the sensitivity of a sub-optimal analysis. We will discuss this next.

## 5 GAN amplification: Relaxing assumption 2

Let us relax assumption 2 of Statement 1 and limit ourselves to a particular, sub-optimal, analysis to be performed using the data. Now we can consider two different versions of the analysis, one which does not use additional GAN-generated data, and one which does. These are represented by $X$ and $X'$ in Figure 3.

In this case, **in principle**, the sub-optimal analysis $X$ can be made more sensitive by using a GAN which can exploit features in the data ignored by the analysis $X$. Furthermore, the worse an analysis $X$ is, the more it stands to gain from using additional GAN-generated data.

This effect was observed in [33], where a binned histogram analysis was found to be improved by using GAN-generated data. In [33], this improvement was captured by an "amplification factor"[9], which can roughly be thought of as the ratio between a) the number of true-simulated events a large GAN-generated dataset ($N \gg N_{\text{train}}$) is worth, and b) the number of training events $N_{\text{train}}$. However, here we will argue that such an amplification will likely not be practically useful.

### 5.1 Leverageable amplification: A contrived example

To see why the amplification may not help realistic analyses, let us first discuss a contrived situation where the amplification does help. Consider a sub-optimal analysis which proceeds as follows:

Given: Some real events and $n$ simulated events.

Step 1: Randomly choose $n/2$ simulated events and discard them. This step serves no purpose other than to make the analysis sub-optimal.

Step 2: Perform an analysis using only the remaining events.

This procedure is clearly sub-optimal, and only uses the information available in half the number of simulated events available to it. One can use a GAN to improve this analysis, by first training it on the $n$ true-simulated events, and then supplying a large number of GAN-generated events to the analysis. In this way, even after discarding half the events provided, the

---

[9]Note that it is unclear whether the advantage offered by a GAN can be captured by a simple amplification factor—different regions of the data domain could enjoy different degrees of "amplification".

analysis can exploit the full information available in the $n$ true-simulated events. In order to reach the same sensitivity without using a GAN, one will have to use $2n$ true-simulated events (half of which will be discarded by the analysis). Thus, in this case, the GAN can amplify the data by a factor of 2.

What makes this amplification useful is the fact that one can *predict* the amplification factor, a priori, using our knowledge of *how* the analysis is sub-optimal. We know that the uncertainties in the final results when using the GAN-generated data, in the context of *this particular analysis*, will be similar to the uncertainties when using $2n$ true-simulated events (assuming that there are no deficiencies in the training of the GAN).

If the amplification factor is not known a priori, then the amplification cannot be leveraged, because one cannot quantify the benefit offered by the GAN. Note that the goal in experiments is to reduce the *reported* uncertainty on measurements or the *reported* upper-limits on signal cross-sections, and not some hypothetical and intangible measures of sensitivity.

## 5.2 The difficulty in realistic scenarios

For a realistic analysis, which is sub-optimal in its usage of the simulated data, it is unclear how to predict the extent to which (and the ways in which) additional GAN-generated data can help the analysis.

To make the discussion concrete, let us consider an analysis performed by analyzing the histogram of an event variable. There is a statistical prescription for drawing error-bars on a histogram with $n$ true-simulated events. More generally, we can quantify the extent to which $n$ true-simulated events will be consistent with the underlying distribution.

Contrarily, we do not know what the error-bars should be on a histogram which uses $n$ GAN-generated events (based on $N_{train}$ training events). In the language of [33], we do not know how many true-simulated events the $N$ GAN-generated events are worth.

In [33], the amplification factor is estimated using the functional form of the underlying theory distribution, by directly calculating the degree of consistency between the underlying distribution and the GAN generated data. However, such a reference handle on the underlying distribution is not available in realistic situations, which is why simulations are needed in the first place.

Another approach to estimating the amplification factor would be to compare the GAN-generated events against more and more true-simulated events, to see where the GAN stops being consistent with the true-simulator. This, of course, would defeat the purpose of using GANs for speedup. The term "amplification factor" might suggest that the amplification provided by a GAN trained on a smaller number of training events, say 500, will be the same as the amplification offered by a GAN trained on a larger number of training events, say 10,000. But this has not been demonstrated in the literature so far.

While practically useful advantage or amplification from using GANs is yet to be demonstrated, we do not claim here that it cannot be done.[10] However, we reiterate that the amplification offered by a GAN depends crucially on the eventual analysis that will use the simulated data. If the analysis is already optimal in its usage of the events available to it, then a GAN cannot improve its sensitivity, as shown in Section 3. In this sense, using GANs to improve sensitivity (if possible), should be considered as a data analysis technique, and not as a simulation technique.[11] The objective of simulators in HEP is to create general purpose datasets

---

[10]It may be possible to estimate the relevant uncertainties (and hence estimate the amplification achieved), using Bayesian neural networks and/or bootstrapping techniques.

[11]Moreover, any increase in sensitivity resulting from improving our current analysis techniques will be no substitute for the typically much larger improvement in sensitivity resulting from a large increase in the amount of available real and simulated data.

that are suitable for a wide range of analyses a practitioner could choose from, and GANs, as discussed so far, cannot satisfy this objective.

# 6 Reconciling our results with earlier studies

The thesis of this paper is in contradiction with the *conclusions* drawn by several previous studies on this topic. However, we believe that there is no evidence in the literature that contradicts our claims. The disconnect between the studies (in particular, their plots and graphs) and the conclusions drawn from them typically arises from a misunderstanding or misrepresentation of what it means for a histogram to be consistent with a distribution (or for two histograms to be consistent with each other). Some common pitfalls we have observed are as follows:

- Histograms of GAN-generated and true-simulated events are sometimes compared with each other only visually, without showing their associated error-bars [25, 27, 29, 32].

- Even if error-bars are shown and the histograms being compared are visibly inconsistent with each other, they are often not acknowledged as such [11, 23]. In several papers [11, 23, 24, 28], standard goodness-of-fit test statistics like the chi-square test statistic are not shown.

- In some papers, where the plots do show the corresponding chi-square test statistic values, large values of the test statistic are not acknowledged as being indicative of poor fits. The conclusions drawn range from "the fit is satisfactory" [30, 31] to more reasonable ones like "the GANs show promise for the future, and can potentially be improved".

These inconsistencies and inadequacies in the literature were the main motivation behind writing this paper. We believe that our point is well known and appreciated by many experts in the Monte Carlo community. Next, we identify the following steps for demonstrating leverageable amplification (when using the GAN to replace the entire MC production pipeline):

1. **Predict the uncertainties.** After training a GAN on some available simulated training data, and producing some number $n_{\text{gen}}$ of GAN-generated events, provide a prescription for estimating the uncertainties (drawing the error-bars) on the histograms[12] built out of the GAN-generated dataset. Note that this should be done without referring to the true-underlying distribution, i.e., without using the functional form of the distribution and without generating additional true-simulated data. In the language of [33], this amounts to *predicting* the amplification factor achieved by the GAN-dataset.

2. **Validate the predicted uncertainties.** Demonstrate that the estimated uncertainties in the GAN-generated dataset are correct (or, at least, not underestimated). This can be done, for example, by showing that the GAN-generated histograms are consistent with the underlying true distributions within the quoted uncertainties, using a statistical test like the chi-square test (calculated using the quoted uncertainties).

3. **Demonstrate improvement.** Demonstrate that the uncertainties when incorporating the GAN-generated dataset are lower than the uncertainties when only using the available true-simulated dataset. This would indicate an improvement in the sensitivity of the analysis resulting from using a GAN.

To the best of our knowledge, a demonstration of this nature does not exist in the current literature.

---

[12]Alternatively, one can also estimate the uncertainty in the *final result* of an analysis performed using the GAN-generated data.

# 7 Potential valid uses of GANs in collider simulations

The event production and reconstruction pipeline employed in high energy physics is a complex one, and addressing the challenge from limited computational resources will require a multifaceted effort. We believe that ML-based generation approaches [51], including GANs, could play a role in the eventual solution. In this section we will identify some applications of GANs in collider simulations, where the usefulness of the GANs is not subverted by the arguments in this paper.

## 7.1 Replacing parts of the Monte Carlo pipeline with GANs

One of the assumptions under which Statement 1 holds is that the GAN will be used to circumvent the entire MC production pipeline. Let us relax this assumption and imagine that the GAN is used to replace a specific stage of the pipeline. In such a situation, using a GAN to accelerate the production pipeline would be warranted if the stage in question is not a major bottleneck on the overall uncertainty from the simulated dataset.

For example, let us consider the case where a GAN is used to map the parton-level event description to reconstructed observables for a particular process (circumventing showering, hadronization, detector simulation, and object reconstruction). One can train a neural network to learn this probabilistic map, using a training dataset comprising of both the parton-level and post-reconstruction descriptions of, say, $N$ events [52]. Such a network can then be used to perform the "partons to detector" transformation[13] on an additional, say, $M$ parton-level events. To be clear, the resulting dataset (consisting of the $N$ true-simulated and $M$ GAN-simulated events) will not contain any more information than the original sample of $N$ true-simulated events and $M$ parton-level events, in accordance with the data processing inequality. The point is, at the data analysis stage, there are not a lot of options for utilizing the additional $M$ parton-level events. The GAN-based simulation simply provides a way to leverage these additional parton-level events at our disposal.

Note that such a GAN-based-simulator will not be an equivalent for the true simulator—if one of the reasons for requiring more simulated events is to estimate the frequency of *rare* detector-induced fakes, then the GAN-based-simulator will not be adequate for the task. While the GAN may not be a statistically equivalent substitute for the corresponding true simulator, it could potentially offer a better trade-off between accuracy and computational/storage resources than other available fast-simulators. It goes without saying that, even in this case, one still needs to estimate the systematic uncertainties in the GAN-generated data (including ML training related uncertainties).

## 7.2 Transfer Learning

Let us continue with the restricted goal of training a GAN to replace a specific stage of the simulation pipeline. Now, the GAN could potentially be trained on external datasets, assuming that the stage under consideration generalizes well across processes.

For example, a GAN for detector simulation can be trained on a variety of datasets containing the detector response for various final state particles, including real calibration data [10]. As a more ambitious example, sufficiently isolated stages of the MC production pipeline could potentially be learned directly from real collision events [53], provided sufficient care is taken to avoid absorbing modeling errors in other stages into the ML model for the stage in consid-

---

[13]Ref. [52] focuses on using the map to perform detector-response unfolding, i.e., probabilistically mapping detector-level events to parton-level events.

eration.[14] In such cases, the information contained in the datasets used to train the generative network (for the specific simulation stage) could be sufficient for the task at hand.

## 7.3 Other miscellaneous applications of GANs

Our arguments do not affect the usage of GANs in situations where a GAN-generated dataset will not be used to yield additional sensitivity in a given analysis. For example, where appropriate, a GAN could be used as a substitute for event resampling techniques like bootstrapping and jackknifing, albeit a more complicated one. GANs can also be used by theorists to produce realistic simulations for their phenomenological studies.

Other possible applications include the use of GANs for unfolding [54,55], denoising [56], event subtraction [57], and model building [58].

## 8 Summary and outlook

In this paper we critically examined the usage of GANs for accelerating the MC production pipeline in particle physics. Based on the data processing inequality, we argued that GAN generated data cannot contain any more information (for the purpose of an experimental analysis) than the data it was trained on. We identified a few scenarios where GANs can nevertheless be usefully utilized for specific tasks and purposes. In any case, one needs to be aware of the limitations of GANs and be ready to put in the necessary work on uncertainty quantification.

While the focus of this paper has been on the usage of GANs for performing collider Monte Carlo, our arguments are also pertinent to other machine-learning-based generative models that learn from simulated training data, including Variational Auto Encoder based approaches. Likewise, our arguments could potentially affect the application of machine learning techniques for simulations outside high energy physics as well [59–62]. We would like to point out that the arguments in this paper are not pertinent to parton-level ML-based generative approaches which learn directly from an oracle which can be queried for the underlying true distribution (matrix-element), instead of learning from data generated under that distribution [35–38, 63–65].

## Acknowledgements

The authors would like to thank D. Acosta, S. Gleyzer, S. Höche, and K. Pedro for useful discussions and feedback.

**Funding information**   This work is supported in part by the United States Department of Energy under Grant No. DE-SC0010296. The work of PS was partially supported by the University of Florida CLAS Dissertation Fellowship funded by the Charles Vincent and Heidi Cole McLaughlin Endowment. PS is partially supported by the U.S. Department of Energy, Office of Science, Office of High Energy Physics QuantISED program under the grants "HEP Machine Learning and Optimization Go Quantum", Award Number 0000240323, and "DOE QuantiSED Consortium QCCFP-QMLQCF", Award Number DE-SC0019219.

This manuscript has been authored by Fermi Research Alliance, LLC under Contract No. DEAC02-07CH11359 with the U.S. Department of Energy, Office of Science, Office of High Energy Physics.

---

[14]It is possible, at least in principle, to even improve certain aspects of our current data generative models (e.g., non-perturbative QCD), by training ML-generative models on real data.

# A  A toy example

In this appendix we shall consider a toy example to demonstrate explicitly the scaling exhibited in Figure 2. For simplicity, we shall consider one-dimensional data for which the true distribution is a Gaussian with standard deviation $\sigma = 1$ and mean $\mu = 0$:

$$f_{true}(x; \mu = 0, \sigma = 1) \equiv \frac{1}{\sqrt{2\pi}} \exp\left(-\frac{(x-\mu)^2}{2\sigma^2}\right) = \frac{1}{\sqrt{2\pi}} \exp\left(-\frac{x^2}{2}\right). \qquad (25)$$

In this toy example, (25) serves as a proxy for the full simulation chain, which is computationally expensive. The standard task would be to start with a small number $n$ of true-simulated events (in this case events produced according to the true distribution (25)) and train a GAN on them which would be able to produce additional data. The training of a GAN is a computationally demanding task, and in practice will not be perfect. In this toy example, however, we know what the ideal GAN should look like — using our knowledge about the true underlying distribution (25), we will simply fit a Gaussian to the training data to obtain the best fit values $\hat{\mu}_n$ and $\hat{\sigma}_n$ from the $n$ training events. This idealized GAN will then simply sample points from a Gaussian distribution with the so found values for $\hat{\mu}_n$ and $\hat{\sigma}_n$.

$$f_{GAN}(x; \hat{\mu}_n, \hat{\sigma}_n) \equiv \frac{1}{\sqrt{2\pi}} \exp\left(-\frac{(x-\hat{\mu}_n)^2}{2\hat{\sigma}_n^2}\right). \qquad (26)$$

In what follows, we shall use this distribution as our proxy for a "perfectly trained GAN", i.e., one which incorporates the prior information about the functional shape of the true model distribution. Even with this advantageous information, $f_{GAN}$ is not a truthful substitute for the true distribution $f_{true}$. Due to the finite statistics in the training data, the derived values for $\mu_n$ and $\sigma_n$ will be slightly different from the true ones, leading to a systematic error in the $f_{GAN}$ generation. This effect is illustrated in Figure 4, where we show results from typical pseudo-experiments in which we train (26) on $n = 100$ events sampled from the true distribution (25). The finitude of the training data may lead to statistical fluctuations in the shape[15] (upper right panel), the mean (lower left panel) or the standard deviation (lower right panel) of the training sample (orange histogram) in comparison to the true distribution (blue dashed line). Only on rare occasions (as in the upper left panel) will the training data resemble the true distribution rather well.

In what follows, we shall focus on the measurement of $\mu$ and the associated statistical errors. To this end, we simulate 100,000 pseudo-experiments with varying number of events $n_{gen} = \{30, 100, 300, 1000\}$, and "measure" the value of $\mu$ as the sample mean. The resulting "measurements" are accumulated in the histograms shown in Figure 5. In the left panel, the $n_{gen}$ events in each pseudo-experiment are sampled from the true distribution (25). This is why, as the statistics increases, the measurement becomes progressively better, and the standard deviation of the measured means scales as $\sigma_{true}/\sqrt{n_{gen}} = 1/\sqrt{n_{gen}}$. In the right panel of Figure 5 we show the corresponding results where the pseudo-experiments are generated with $f_{GAN}$, after first fitting for $\mu_n$ and $\sigma_n$ to $n = 100$ training events (we use a *different* training sample for each pseudo-experiment). This time, however, the increase in statistics does *not* lead to an associated improvement in precision, and in the limit of $n_{gen} \to \infty$ the distribution asymptotes to the green histogram in the left panel for $n_{gen} = 100$. This can be understood as follows. There are two types of errors entering the results in the right panel of Figure 5: first, the statistical error from the training sample, which is subsequently inherited as a systematic error $\sigma_{syst}$ in the GAN-generated data; and second, the statistical error $\sigma_{stat} \sim 1/\sqrt{n_{gen}}$. We

---

[15]However, since we are fitting a Gaussian to the data, this type of fluctuation does not degrade the quality of the data generated by $f_{GAN}$.

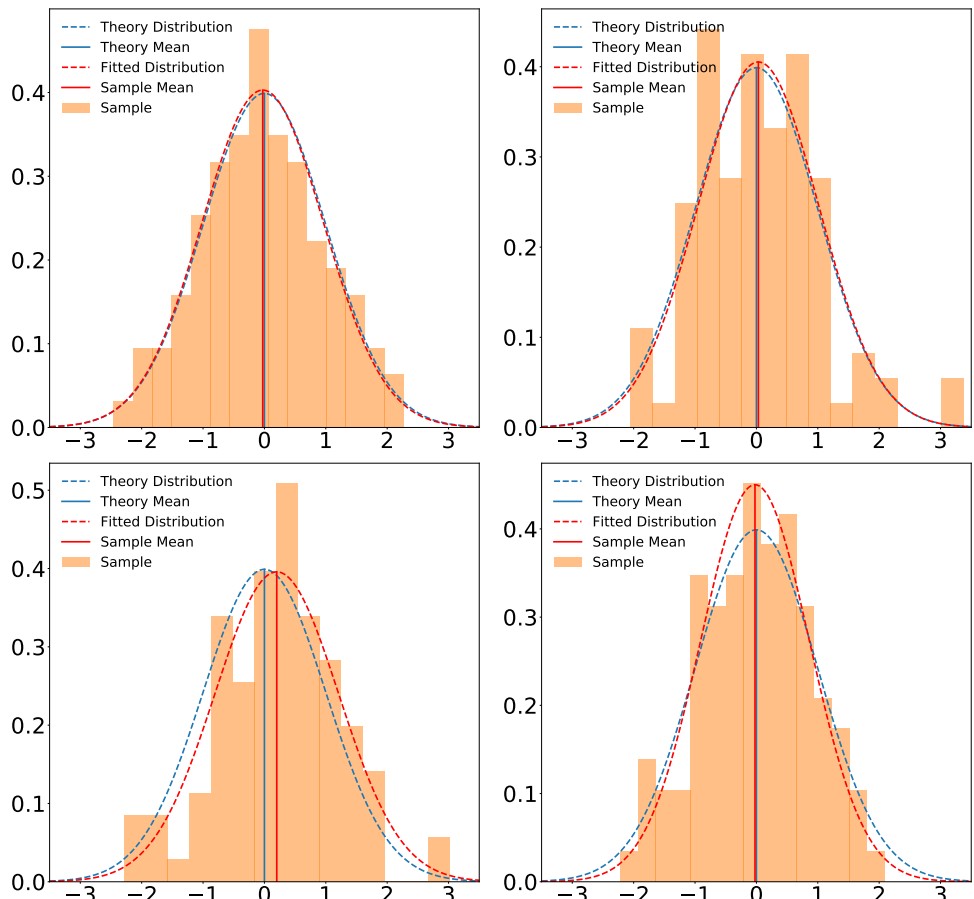

Figure 4: Results from typical pseudo-experiments with $n = 100$ events each, drawn from the true distribution (25) (the blue dashed line). Each panel shows an orange-shaded normalized histogram of the sampled data, as well as a Gaussian fit to it (red dashed line) and the derived sample mean $\mu_n$ (red solid line). The four pseudo-experiments depicted here illustrate different levels of agreement between the sampled data and the underlying theoretical distribution. In the upper row, the fit to the data has similar mean $\mu_n$ and standard deviation $\sigma_n$ as the theory distribution, but in the left (right) panel in addition it has a good (bad) $\chi^2$. In the lower left (lower right) panel the mean (standard deviation) of the fit is noticeably different from the true value.

have verified that adding these two uncertainties in quadrature predicts the total uncertainties observed in the right panel of Figure 5. The evolution of these uncertainties with the sample size $n_{gen}$ is shown in Fig. 6, which is the quantitative analogue of Fig. 2.

Figure 7 provides another point of view which could be helpful in understanding the results from Figure 5 and Figure 6. The main problem is that the sample mean observed in the training data is *not* necessarily equal to the true value of $\mu$. This is illustrated in the upper panel of Figure 7, which shows the distribution of sample means for 300,000 pseudo-experiments, each consisting of $n_{gen} = 100$ points sampled from the true distribution (25). We observe that the distribution is centered at 0 (the true value of $\mu$ in our toy example) and has a standard deviation of approximately 0.1, i.e., $\sigma_{\text{true}}/\sqrt{n_{\text{gen}}}$. When we train a "GAN" with $n = 100$, the training sample itself will have a mean which will be distributed as in the top panel of Figure 7, and this uncertainty will be inherited by the $f_{\text{GAN}}$ generator. For illustration we have picked three representative training datasets whose means are denoted with the red dots. For each

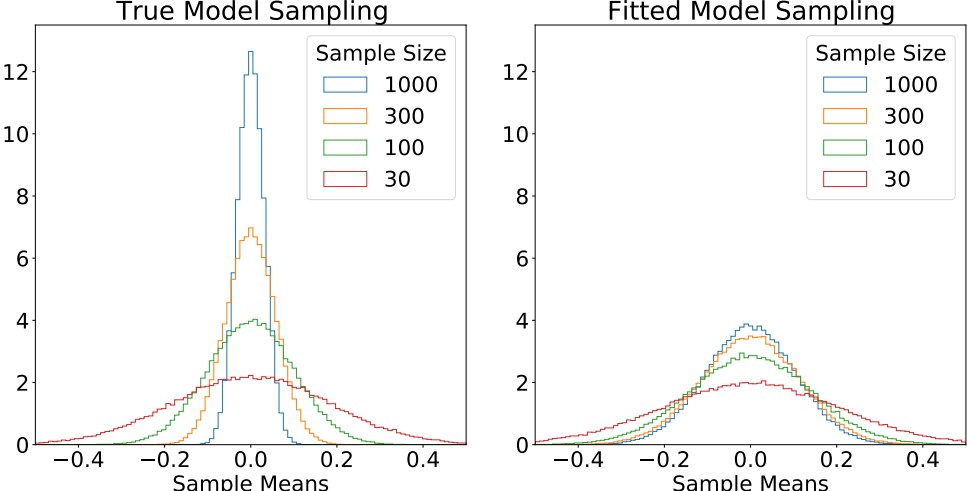

Figure 5: Distribution of the sample means in 100,000 pseudo-experiments for different sample sizes, i.e., number of events in a given pseudo-experiment. In the left panel the events are drawn from the true theory distribution (25), while in the right panel the events are sampled from the $f_{\text{GAN}}$ distribution (26) trained on $n = 100$ events from the true distribution (25).

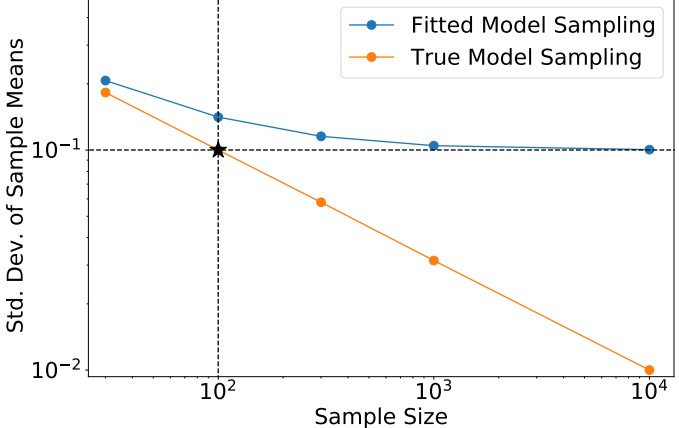

Figure 6: Evolution of the standard deviation of the sample means with respect to the sample size. The orange line corresponds to the case where events are sampled from the true distribution as in the left panel of Figure 5. The blue line corresponds to the case from the right panel of Figure 5, where events are generated from $f_{\text{GAN}}$ fitted to $n = 100$ events sampled from the true distribution.

of those samples, we then train the $f_{\text{GAN}}$ generator, and use it to produce increasingly larger datasets of $n_{gen}$ events. The lower panel in Figure 7 shows the mean (black dashed line) and standard deviation (red shaded funnels) as a function of the sample size $n_{gen}$ estimated from 1,000 pseudo-experiments for each of the three $f_{\text{GAN}}$ realizations. As the statistics increase, the funnels eventually converge to the original means, as indicated by the three red points. However, note that the asymptotic values are not correlated with the true value of $\mu = 0$, but instead with the starting values in the training data (the red dots in the top panel). As a result, the distribution of asymptotic values for GAN-generated data (the faint blue dots in the bottom panel) simply mimics the distribution of the sample means in the training data (the blue histogram in the top panel).

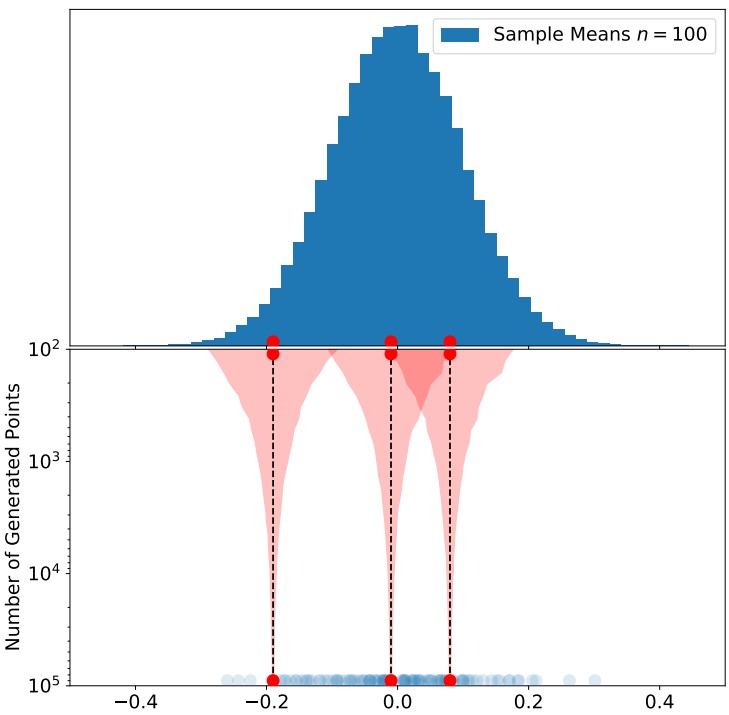

Figure 7: The blue histogram in the top panel depicts the distribution of sample means of pseudo-experiments consisting of $n_{gen} = 100$ points sampled from the true distribution (25). This distribution is centered at 0 and has a standard deviation of approximately 0.1. The red dots in the top panel depict three representative pseudo-experiments, which are then used to train $f_{\text{GAN}}$. The lower panel shows the mean (black dashed line) and standard deviation (red shaded funnels) as a function of the sample size $n_{gen}$ estimated from 1,000 pseudo-experiments for each of the three $f_{\text{GAN}}$ realizations. The funnels eventually converge to the original means, as indicated by the three red points. The faint blue points in the lower panel show 100 more examples of this procedure, and are distributed according to the original blue histogram in the upper panel.

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
