# Peer review of "Uncertainties associated with GAN-generated datasets in high energy physics"

_SciPost Physics, doi:SciPost Phys. 12, 104 (2022)_

## Round 2 · Referee Report · Anonymous (Referee 1) · 2020-7-1

Strengths

The paper is asking an extremely important question, relevant not only for particle physics applications of generative networks, but also for broader applications of machine learning.

Weaknesses

The argument given in Sec.2 is neither mathematically or formally backed up, nor is it illustrated by an analysis. More than that, looking at broader applications of machine learning I would argue that it is most likely wrong. Specifically: 1- The sentence `This implies that the model (parameter) discriminating power...' comes out of nowhere. Why is this implied? 2- The example given below is not correct. A network can interpolate and it can learn an (approximate) functional form, because a neural network is nothing but a learned function. Sampling from a network can improve the analysis, even it those additional events are not a fully adequate replacement for the same number of properly sampled points. Where am I wrong? 3- In Sec.2.2 it sounds like the authors are complaining correctly that GAN analyses typically fail to address the issue of robustness and information content. This is true, I completely agree, but this does not mean that it could not be shown; 4- Making a cynical argument like at the end of Sec.2.2 really requires to be right, otherwise it is irritating and not appropriate; 5- I am not sure why the authors only discuss GANs and not other generative networks. Is this meant as an indication that the argument does not hold generally? For instance event generation is done with GANs the same way it is done with VAEs.

Report

Please note that I am not saying that I can solve the problem the authors address. The issue I have is that I see plenty of evidence that either their general claim is wrong, or it is so specific that most applications of generative networks will end up on the extensive list of exceptions. The latter case would be interesting formally, but it needs a solid proof, separating the good use of generative networks from the bad use.

Requested changes

Please provide a formal and solid proof for the claims in Sec.2. And I am sorry, but my guess is that this will not be possible, because the evidence I have seen in the literature points in the other direction.

  • validity: poor
  • significance: poor
  • originality: poor
  • clarity: low
  • formatting: excellent
  • grammar: excellent

Author:  Prasanth Shyamsundar  on 2021-06-25  [id 1524]

(in reply to Report 1 on 2020-07-01)
Category:
answer to question
reply to objection

We thank the referee for their comments and suggestions.
Addressing the list of perceived weaknesses: 1. We thank the referee for this comment. We have converted the sentence under question into a theorem in Section 2, and supported it with mathematical demonstrations in Section 3, a proof in Section 4, and a toy example in Appendix A (all newly added). 2. We thank the referee for this comment. This is a common argument used in favor of GANs. We address this in Section 3.4 and Section 5 (both newly added). 3. We support our claims regarding the limitations of GANs rigorously in Sections 3 and 4 and Appendix A. 4. The argument under question is correct, and to better get the point across, we have rephrased the wording (now at the end of Section 4) and added Figure 1, where the parallels between GANning simulated data and GANning real collider data are explicit. 5. Our arguments do apply to other generative models like VAEs. Our focus on GANs was dictated by their prevalent usage in the literature. We have modified our summary section to indicate this.

Regarding the criticism in the report: We believe that there is no evidence in the literature that our claims are wrong. We discuss this in the newly added Section 6, which reconciles our results with those in the literature. In the revised version, we have extensively discussed examples of both good and bad usages of GANs, as requested.

We have also addressed the requested change by supporting our claims rigorously in Section 3 and 4. The paper has been almost completely rewritten, and now stands at ~27 pages as opposed to the previous version at ~11 pages.

Regards,
Prasanth

---

## Round 2 · Referee Report · Anonymous (Referee 2) · 2020-7-7

Strengths

Provides a welcome critique of sometimes over-reaching claims for generative methods, particularly as applied to event generation, with many sensible comments on the propagation of statistical unceratinties through analysis chains.

Weaknesses

  1. The overall contention statistical convergence to the true asymptotic behaviour cannot achieved by generative methods from a low-statistics input sample is obvious, although I would agree that this is sometimes not obvious from the literature on such techniques.

  2. The end of Section 1 says that what follows will be "presented carefully", but to my surprise the argument made is not analytical, but polemical, via analogy and intuition, and does not add much to the statement of Section 1. I actually rather enjoyed the presentation, but the absence of formal proof weakens the case for publication, especially given that the core insight is not novel.

  3. The bold statement on p3 seems to me to itself be an over-reach, and is a key example of where a formal approach would greatly improve the strength of the argument. I suspect that it is technically correct for a relatively narrow definition of "GAN-generated", but the informal nature of the presentation means that definition is not fully clear. In particular, a process limited by detector simulation costs rather than event generation evidently could achieve improved accuracy for parameter inference, provided the statistics of the detector response training (as opposed to the physics MC samples) is not a leading systematic. On the other hand, if the fundamental process-modelling statistics are a limitation, the input to GAN training itself has statistical uncertainties which cannot obviously be reduced by (effectively) bootstrapping. Such gaps in the argument are in fact nicely summarised in the caveats of Section 3, where the difference between in-principle and pragmatic limitations is made clear.

Report

Apologies for this late review, due to the new global circumstances. I enjoyed the short paper, but found it lacking in rigour which undermines the point being made (which I think is very sensible if not 100% watertight, but well known in at least the event generation community).

As a sensible commentary on not getting too excited about AI methods to the extent of forgetting basic statistics, it is rather good, but I am not convinced that such a polemic justifies journal publication. And given the lack of rigour in the central section, I think it would be more effective as messaging to compress Sections 1 and 2 into a single succinct summary of the argument, followed by the Caveats making clear that the perfect need not be the enemy of the good.

Requested changes

  1. Make a more rigorous argument in Section 2, clarifying through formalism the scenarios and uncertainty classes being referred to.

  2. Reduce repetition of the well-known main argument... but I suspect this would happen anyway by formalising Section 2.

  • validity: good
  • significance: low
  • originality: low
  • clarity: good
  • formatting: excellent
  • grammar: excellent

Author:  Prasanth Shyamsundar  on 2021-06-25  [id 1525]

(in reply to Report 2 on 2020-07-07)

We thank the referee for their comments and suggestions.
Addressing the list of perceived weaknesses: 1. Our focus in this paper is not just on convergence to true-asymptotic behavior, but on discrepancy between the generative models and true distribution even at finite statistics. In this sense, our main arguments are not trivial. 2. We thank the referee for the suggestions. We now present our argument in the form of a theorem in section 2 and information theoretic demonstrations in section 3 (both newly added in this version). 3. We have made the presentation of our arguments more formal throughout the paper to address these concerns.

We have also addressed both the requested changes with an almost complete rewrite of the paper, which now stands at ~27 pages as opposed to the previous version at ~11 pages.

Regards,
Prasanth

---

## Round 3 · Referee Report · Anonymous (Referee 1) · 2021-7-7

Report

First of all, I would like to thank the authors for being much more specific than before. I think their point is quite clear now, even though their separation of generation and analysis appears a little ad-hoc and I am not sure how it is related to the paper title. Essentially, the authors now say that generative networks are not MC generators, which might not be all that new or insightful, but it is correct. Any kind of improvement beyond standard MC is dubbed analysis, which is weird, given that it also applies to modern methods like event reweighting etc. Only one remaining comment, given that times are moving fast, please comment on the recent papers on network-based unweighting. Those advertize learning a phase space density on weighted events and sampling unweighted events, maybe with a correction to the true density via post-processing, so I am curious how they fit into this specific scheme.
  • validity: -
  • significance: -
  • originality: -
  • clarity: -
  • formatting: -
  • grammar: -

Author:  Prasanth Shyamsundar  on 2022-02-17  [id 2218]

(in reply to Report 1 on 2021-07-07)
Category:
remark
answer to question

  • "First of all, I would like to thank the authors for being much more specific than before. I think their point is quite clear now, even though their separation of generation and analysis appears a little ad-hoc and I am not sure how it is related to the paper title. Essentially, the authors now say that generative networks are not MC generators, which might not be all that new or insightful, but it is correct. Any kind of improvement beyond standard MC is dubbed analysis, which is weird, given that it also applies to modern methods like event reweighting etc."

We thank the referee for the feedback. We disagree that the distinction between generation and analysis is ad-hoc. A collider analysis can be made more sensitive by 1) increasing the amount of real/simulated data; and/or 2) by improving the techniques used to analyze the data. Although the end result (i.e., improved sensitivity) is the same, neither approach is a substitute for the other. We believe that it is misleading to claim to be solving one problem (insufficient simulation statistics) by addressing the other (improving the analysis). Furthermore, as we showed in the paper, one cannot use GAN-generated data in the same way as one would use the standard MC-generated data. It is in that sense that the separation between generation and analysis is justified and relevant for the discussion in this paper.

  • "Only one remaining comment, given that times are moving fast, please comment on the recent papers on network-based unweighting. Those advertize learning a phase space density on weighted events and sampling unweighted events, maybe with a correction to the true density via post-processing, so I am curious how they fit into this specific scheme."

We thank the referee for the suggestion. ML-based event weighting, unweighting, and reweighting strategies is an exciting recent development. In analyses which use ML-weighted datasets, it is absolutely critical to account for uncertainties resulting from finiteness of the training data. We are currently working on a separate paper addressing this topic.

---

## Round 3 · Referee Report · Andy Buckley (Referee 3) · 2021-8-19

Strengths

  1. Highlighting a well-known but not always appreciated fundamental limitation in use of generative models to address statistical limitations in full MC event-generation chains.

  2. Pedagogically useful and maybe novel applications of the data-processing inequality to show the intuitive result that GANs trained on a model-derived dataset cannot increase information about the model via a GAN-generated dataset.

  3. Useful contextual discussion of claims for GAN interpolation as inference, and proposed procedural improvements for reports on GAN-based HEP generator performance.

Weaknesses

  1. The central theorem is shown in its "proof" to be phrased tautologically and hence add nothing to the discussion (which is qualitative, but valid and useful)

  2. The main thrust of argument - that GANs do not improve statistical convergence to the true model - is already well-known and appreciated among the MC generator community (and I'm sure more widely)

  3. Much of the discussion is nicely phrased, but seems more verbose than necessary: some simple things are explained to the point of confusion, and the conclusions/recommendations from each section's argument are not always clear.

Report

A well-presented paper, but the main message is already well-known based on knowledge or intuition of the data-processing inequality, as the authors admit in Section 6. The discussions are interesting and detailed, if rather verbose, and as such I feel it contributes well to the discourse around where generative models can add value, and how to quantify it. However, I do not see the point in spending pages on "proving" a tautological theorem: "allowing GANs in an analysis chain cannot improve on the best performance if GANs were not previously forbidden" is not an important result and hence distracts from the useful comments elsewhere.

What is left is pedagogically useful, particularly the explicit demonstrations of the data-processing inequality as applied to Markovian generative models, and I am glad it is in the public literature, but I am not sure it is suitable for publication as a novel article as opposed to a commentary or review. I do not think it meets the stringent acceptance conditions of SciPost Physics, but could be suitable for SciPost Physics Core.

Detailed comments:

• p3: the statement of the theorem is vague as to the meaning of "discriminating power". While an up-front statement of the result is welcome, postponing the detail of what it actually means until the proof is unhelpful. I would add that this theorem is unsurprising to all I am aware of working in this area (to the point of having been assumed true), although I'm not aware of a previous explicit proof. Indeed, the assumption made in this proof that the GAN will replace the entire chain from the fundamental process onward is the very reason that MC-generator authors have been critical of generation GANs, and encouraged focusing such methods on less critical downstream simulation, cf. Section 7.

• p5: this seems to be a wordy way of saying that model inferences require a good estimation of the mean rates of bin/signal-region population by the model, and the estimates of these mean rates have an undesirable uncertainty due to finite MC statistics. It is then clear that the accuracy of such estimates are set by the training sample, as any generative model based on it has no information with which to systematically improve on the estimate. As later commented in Section 6, this fact is well appreciated in the MC community.

• eq 2: this analysis is sound, but it gives the impression that the fully specified likelihood ratio is the real quantity of interest, while in practice the intermediate states are not of interest. If the likelihood ratio were considered on partially specified states, e.g. the ratio of P(D_GAN | theta_i) between i=1 and i=0 -- the probability of the same GAN-generated dataset having resulted from two different input models -- then the summation over intermediate "paths" makes the cancellation less obvious and hence the information theoretic proofs more interesting. The first proof (on the mutual information) I think does not rely on the likelihood ratio or score, and so the structure would maybe be better with these results postponed to where they are used in the KL and Fisher proofs.

• 3.3: while nicely explained, I think most readers probably approach this backward, being familiar with the fact that generation by sampling from training data will not improve convergence to the true model, but not having seen the preceding proofs.

• 3.4: eq. 24 and the equivalent points above is an assumption, not always true. The point that interpolations or smoothings (including GANs) are neither unqiue nor a priori closer to the truth than the unsmoothed dataset is well made, however. An additional issue, mentioned in the context of Section 7 but not here, is with emergent features due to low-rate physics effects that are likely to be completely unrepresented in the training sample: there is no reason at all to expect that such features can be "interpolated" into existence by the latent forms in the GAN machinery.

• Section 4: the "proof" is rather a neat thought experiment, but the meaning of the X and X' in Fig 3 are not clear and do not seem to correspond to anything in the text (the proof is not symbolic). However, my fundamental concern is that the proof is really a statement that the theorem's conditions are tautological. This is not a good thing: it effectively removes it from the discussion entirely, leaving the following argument for the limitations of GANs to depend entirely on subjective judgements. As a result I think you have shown Theorem 1 to be a red herring that adds little if anything to the clarity of the paper's core message, and quite possibly detracts by introducing an irrelevant complications. I think the information-theoretic statements are far stronger, if "obvious" arguments for the fundamental limitations of GANs (or at least those which attempt to cover the fundamental theory model).

• Section 6: "several other studies" absolutely needs a set of corresponding citations. It would also assist greatly, despite seeming perhaps aggressive, if the critiques in the list below also attached citations to clarify which studies are being referred to. To not cite in this criticism section is guilty of the worse crime of passive aggression! I say this as someone sympathetic with your implied criticism that ML-applications papers sometimes do not consider that the paradigm could be imperfect, and excuse suboptimal performance with banalities. The recommendations for publishing GAN performance are good but could be improved by suggesting explicit methods for estimating GAN uncertainties -- an obvious one is to train many GANs based on randomly chosen training sets from a larger total set of training events, and calculate the variances in GAN'd results over the set of GANs, but there are maybe smarter ways.

• Section 7: I may be unfamiliar with the ways envisaged for use of simulation+reco GANs, but had not imagined use of a mixed full-sim/reco and GAN'ed event set. Use of GANs to replace finite pre-calculated shower libraries and similar seems a more obvious approach, as taken by e.g. CaloGAN (https://arxiv.org/abs/1712.10321) . This, however, has the huge caveat that additional methods are needed to adapt the shower generation to the new, continuous parameter space of truth-particle/jet kinematic phase-space. Maybe worth mentioning or focusing on this approach, rather than defeating a straw-man -- if the "N - M" approach is in active use, a citation would be appropriate. There are many more distinctions between the simulation+reco step as compared to fundamental-model sampling, probably too many to mention: the relative regularity and central-limit nature of detectors, the (un)importance of rare detector phenomena, the roles of cleaning cuts and object calibrations, and the fundamental distinction in importance between accurate inference of detector nuisance parameters vs fundamental-physics parameters. Whether GANs are an appropriate replacement for elements in the post-generation chain seems to depend a lot on what they are to be used for, and whether it would depend on the existence of rare configurations in the training sample.

Requested changes

  1. Given the tautology, I do not see that Theorem 1 adds any value to the paper. I would suggest removing it except perhaps for including the observation in textual discussion, and focusing more clearly on the key issue as nicely clarified by the information-theoretic derivations.

  2. In the Report section I suggest some possible, optional, improvements to the information theory presentation to make the flow of argument easier to follow.

  3. In the Section 6 critiques of earlier studies, it is crucial to add citations to the "other studies" as appropriate, otherwise the critique is flogging a straw man.

  4. Optionally expand on the issues for post-generation uses of GANs to address statistical/CPU bottlenecks. This seems a more realistic use of GANs due to the existing awareness of the issues for physics-model sampling, and I feel there is much interesting discussion lying in the distinction between the two modes.

  • validity: good
  • significance: ok
  • originality: ok
  • clarity: high
  • formatting: perfect
  • grammar: perfect

Author:  Prasanth Shyamsundar  on 2022-02-17  [id 2217]

(in reply to Report 2 by Andy Buckley on 2021-08-19)
Category:
remark
answer to question

  • "Given the tautology, I do not see that Theorem 1 adds any value to the paper. I would suggest removing it except perhaps for including the observation in textual discussion, and focusing more clearly on the key issue as nicely clarified by the information-theoretic derivations."

We thank the referee for the feedback. This is one of several conflicting recommendations in this regard which we have received from referees. The original versions of the paper did not have a formal "theorem", but its content was discussed in the text. However, several referees were skeptical of its validity, which forced us to formalize the statement and prove it. In the new version we have renamed it as simply a "statement".

  • "In the Report section I suggest some possible, optional, improvements to the information theory presentation to make the flow of argument easier to follow."

We thank the referee for the numerous optional suggestions. We went through the comments and optional suggestions in the report section and in the revised version incorporated some of them as follows:

  • "p3: the statement of the theorem is vague as to the meaning of "discriminating power". While an up-front statement of the result is welcome, postponing the detail of what it actually means until the proof is unhelpful. I would add that this theorem is unsurprising to all I am aware of working in this area (to the point of having been assumed true), although I'm not aware of a previous explicit proof. Indeed, the assumption made in this proof that the GAN will replace the entire chain from the fundamental process onward is the very reason that MC-generator authors have been critical of generation GANs, and encouraged focusing such methods on less critical downstream simulation, cf. Section 7."

    In the revised version we have clarified that the statement holds for any agreed-upon evaluation metric capturing the discriminating power. We also agree with the referee that there is a portion of the community (us included) to whom the theorem is unsurprising, but, as the referee points out, there is also a substantial literature on generation GANs which this paper is addressing. The goal of this paper is to formalize the criticism the referee is alluding to, and place it in the literature.

  • "Section 4: the "proof" is rather a neat thought experiment, but the meaning of the X and X' in Fig 3 are not clear and do not seem to correspond to anything in the text (the proof is not symbolic). However, my fundamental concern is that the proof is really a statement that the theorem's conditions are tautological. This is not a good thing: it effectively removes it from the discussion entirely, leaving the following argument for the limitations of GANs to depend entirely on subjective judgements. As a result I think you have shown Theorem 1 to be a red herring that adds little if anything to the clarity of the paper's core message, and quite possibly detracts by introducing an irrelevant complications. I think the information-theoretic statements are far stronger, if "obvious" arguments for the fundamental limitations of GANs (or at least those which attempt to cover the fundamental theory model)."

    Such thought-experiment-based proofs for arguably obvious results are not uncommon in science, e.g., the "no free lunch" theorem for search and optimization problems. Any application of GANs that falls within the assumptions of our statement, is objectively limited by it.

  • "Section 6: "several other studies" absolutely needs a set of corresponding citations. It would also assist greatly, despite seeming perhaps aggressive, if the critiques in the list below also attached citations to clarify which studies are being referred to. To not cite in this criticism section is guilty of the worse crime of passive aggression! I say this as someone sympathetic with your implied criticism that ML-applications papers sometimes do not consider that the paradigm could be imperfect, and excuse suboptimal performance with banalities. The recommendations for publishing GAN performance are good but could be improved by suggesting explicit methods for estimating GAN uncertainties -- an obvious one is to train many GANs based on randomly chosen training sets from a larger total set of training events, and calculate the variances in GAN'd results over the set of GANs, but there are maybe smarter ways."

    We followed the referee's advice and added the corresponding citations to section 6.

  • "Section 7: I may be unfamiliar with the ways envisaged for use of simulation+reco GANs, but had not imagined use of a mixed full-sim/reco and GAN'ed event set. Use of GANs to replace finite pre-calculated shower libraries and similar seems a more obvious approach, as taken by e.g. CaloGAN (https://arxiv.org/abs/1712.10321) . This, however, has the huge caveat that additional methods are needed to adapt the shower generation to the new, continuous parameter space of truth-particle/jet kinematic phase-space. Maybe worth mentioning or focusing on this approach, rather than defeating a straw-man -- if the "N - M" approach is in active use, a citation would be appropriate. There are many more distinctions between the simulation+reco step as compared to fundamental-model sampling, probably too many to mention: the relative regularity and central-limit nature of detectors, the (un)importance of rare detector phenomena, the roles of cleaning cuts and object calibrations, and the fundamental distinction in importance between accurate inference of detector nuisance parameters vs fundamental-physics parameters. Whether GANs are an appropriate replacement for elements in the post-generation chain seems to depend a lot on what they are to be used for, and whether it would depend on the existence of rare configurations in the training sample."

    We thank the referee for the feedback. The purpose of this section was not to critize any particular usage of GANs, but rather to indicate a potentially valid usage of GANs. We have changed the title of section 7 from "Potential uses of GANs in collider simulations" to "Potential valid uses of GANs in collider simulations". We have also rewritten parts of section 7.1 to be clearer.

  • "In the Section 6 critiques of earlier studies, it is crucial to add citations to the "other studies" as appropriate, otherwise the critique is flogging a straw man."

We thank the referee for the feedback. We have now included proper references in section 6.

  • "Optionally expand on the issues for post-generation uses of GANs to address statistical/CPU bottlenecks. This seems a more realistic use of GANs due to the existing awareness of the issues for physics-model sampling, and I feel there is much interesting discussion lying in the distinction between the two modes."

We thank the referee for highlighting those issues, which are beyond the scope of this paper and we may revisit in a future paper.

---

## Round 3 · Referee Report · Anonymous (Referee 4) · 2021-9-8

Strengths

1 - The manuscript points out a logical flaw in some high-energy physics applications of GANs. The problem is very basic in nature, but its consequences are often underappreciated.

Weaknesses

1 - The paper is, at times, too provocative. While the statements are factually correct, they should be presented in a more productive way.

Report

The manuscript deals with the question whether machine learning can be used to increase the statistical significance of simulated data sets based on a known underlying physics model. Using basic information theoretical arguments, the authors find this not to be the case, and conclude that various existing works on this topic may draw the wrong conclusions.

While the analysis of the problem is correct, I find the presentation rather problematic. The authors should consider rephrasing some of their statements, in particular

  • The last paragraph of Sec.3: This comment is polemic, but it can in fact be used to bolster the case for the manuscript if it is taken as an explicit example of how not to use a GAN. It should be rephrased and moved from Sec.3 to the introduction.
  • I would caution against calling "Theorem 1" a theorem. The paper in its present form has drawn attention for the wrong reason, with several theorists criticizing its lack of content. This could be avoided by not pretending there to be significant math behind the theorem, but instead giving very explicit examples of unintended consequences when GANs are used to enhance statistical significance.
  • I would therefore also like to see App.A moved into the main body of the text. This explicit example can be utilized to make the argument in a clear and convincing manner. Figure 6 in particular will be helpful for this. It should be plotted on a log-log scale, the same as Fig.2. Figure 2 itself could be removed.
  • The authors should make it clear that the main problem is not the usage of GANs, but the lack of error propagation. If the statistical uncertainty of the training data was quoted as a systematic uncertainty on the final prediction, there would not be a problem in the first place.

Requested changes

See report above

  • validity: ok
  • significance: good
  • originality: ok
  • clarity: low
  • formatting: good
  • grammar: good

Author:  Prasanth Shyamsundar  on 2022-02-17  [id 2216]

(in reply to Report 3 on 2021-09-08)
Category:
remark
answer to question

  • "The last paragraph of Sec.3: This comment is polemic, but it can in fact be used to bolster the case for the manuscript if it is taken as an explicit example of how not to use a GAN. It should be rephrased and moved from Sec.~3 to the introduction."

We thank the referee for the suggestion. As requested, we rephrased and moved that paragraph to the introduction.

  • "I would caution against calling "Theorem 1" a theorem. The paper in its present form has drawn attention for the wrong reason, with several theorists criticizing its lack of content. This could be avoided by not pretending there to be significant math behind the theorem, but instead giving very explicit examples of unintended consequences when GANs are used to enhance statistical significance."

This is one of several conflicting recommendations in this regard which we have received from referees in the past. The original versions of the paper did not have a formal "theorem", but its content was discussed in the text. However, several referees were skeptical of its validity, which forced us to formalize the statement and prove it in the revised version. In the new version we have renamed it as simply a "statement".

  • "I would therefore also like to see App.A moved into the main body of the text. This explicit example can be utilized to make the argument in a clear and convincing manner. Figure 6 in particular will be helpful for this. It should be plotted on a log-log scale, the same as Fig.2. Figure 2 itself could be removed."

We thank the referee for the feedback. Figure 2 serves a useful purpose because it is generic and conveys the message that our statement is more universally applicable than a single toy example would suggest. Figure 6 is also useful, as it quantitatively backs up Figure 2 with a concrete example. Following the suggestion of the referee, in the revised version Figure 6 is re-plotted on a log-log scale.

  • "The authors should make it clear that the main problem is not the usage of GANs, but the lack of error propagation. If the statistical uncertainty of the training data was quoted as a systematic uncertainty on the final prediction, there would not be a problem in the first place."

We thank the referee for the feedback. We agree that propagating all the errors from the usage of machine learning will make the resulting analysis correct. However, when the stated reason for using a GAN is to reduce an uncertainty that will (have to) be propagated under a different name anyway, then there is a more fundamental problem with the approach (and not just an error-propagation issue).

---

## Round 3 · List of Changes

The manuscript has undergone several major changes. We have made our arguments more quantitative by * Formulating the main argument as a theorem in Section 2 and proving it in Section 4. * Providing three different information theoretic demonstrations (using mutual information, Fisher information, and KL divergence) in Section 3, which show that the GAN generated dataset cannot contain any more information than the training dataset it is based on. * Providing a toy example in an appendix which demonstrates our claims. In addition, we also * Address the argument in favor of GANs based on their ability to be good function approximators (Section 3.4) * Discuss a recent work in the literature which suggested the possibility of amplifying datasets using GANs (Section 5) * Reconcile our results with those of earlier studies, providing an explanation for the seeming incompatibility of our claims with the evidence in the literature (Section 6) * Identify some applications of GANs that are not subverted by the arguments presented in the paper (Section 7)

While we have improved the presentation of our arguments in this version and added new material, our claims (along the caveats) presented in the previous version remain unchanged and have not been weakened.

---

## Round 4 · List of Changes

We have made a number of minor changes in response to the referees' comments. These changes are described in our responses to the individual referee reports on the previous version of this manuscript.

---

## Editorial Decision

published